# Hypothalamic Actions of SIRT1 and SIRT6 on Energy Balance

**DOI:** 10.3390/ijms22031430

**Published:** 2021-01-31

**Authors:** Mar Quiñones, Eva Martínez-Grobas, Johan Fernø, Raquel Pérez-Lois, Luisa María Seoane, Omar Al Massadi

**Affiliations:** 1Department of Physiology, CIMUS, University of Santiago de Compostela-Instituto de Investigación Sanitaria de Santiago de Compostela, 15782 Santiago de Compostela, Spain; eva.martinez.grobas@rai.usc.es; 2CIBER Fisiopatología de la Obesidad y Nutrición (CIBERobn), 15706 Santiago de Compostela, Spain; luisamaria.seoane@usc.es; 3Hormone Laboratory, Department of Medical Biochemistry and Pharmacology, Haukeland University Hospital, N-5021 Bergen, Norway; johan.ferno@uib.no; 4Instituto de Investigación Sanitaria de Santiago de Compostela, Complexo Hospitalario Universitario de Santiago (CHUS/SERGAS), Travesía da Choupana s/n, 15706 Santiago de Compostela, Spain; raquelpl.93@gmail.com

**Keywords:** SIRT1, SIRT6, energy balance, food intake, body weight, adiposity, type 2 diabetes, obesity

## Abstract

Sirtuins are NAD+ dependent deacetylases that regulate a large number of physiological processes. These enzymes are highly conserved and act as energy sensors to coordinate different metabolic responses in a controlled manner. At present, seven mammalian sirtuins (SIRT 1-7) have been identified, with SIRT1 and SIRT6 shown to exert their metabolic actions in the hypothalamus, both with crucial roles in eliciting responses to dampen metabolic complications associated with obesity. Therefore, our aim is to compile the current understanding on the role of SIRT1 and SIRT6 in the hypothalamus, especially highlighting their actions on the control of energy balance.

## 1. Introduction

### 1.1. Obesity

The incidence of obesity has reached pandemic proportions and become a major cause of various social, economic and health problems in our society [1]. In fact, worldwide obesity has nearly tripled since 1975 and the World Health Organization has declared that overweight is among the top ten conditions of major health risk in the world and among the top five in developed countries [2]. Obesity is defined as a state on excess of fat produced as a consequence of a high energy intake and a low energy expenditure that leads to serious health problems [3,4]. While food intake is tightly regulated by homeostatic and hedonic drives [5], changes in energy expenditure can be explained by differences in basal metabolism (energy expended to keep the body functioning at rest), physical activity and adaptive thermogenesis [6]. Therefore, obesity is a complex and multifactorial disease involving genetic, biological but also behavioral factors.

The first line of treatment of obese people is to follow a healthy diet and do regular exercise. However, these guidelines are difficult to adhere to for the obese population and most people with obesity that lose weight will regain it over time, ultimately with a higher body weight than before starting the weight loss [7,8]. Bariatric surgery is currently the most efficient method for long-term weight loss. However, this is a costly and invasive intervention not without risk and side-effects, and is only recommended to a small proportion of morbidly obese people [9]. The currently available pharmacological treatments to combat obesity have limited effectiveness and more efficient drugs are urgently needed. Therefore, it is necessary to ascertain in depth the molecular mechanisms underlying this disease and to identify novel therapeutic targets for treatment.

### 1.2. The Hypothalamus as the Master Regulator of Energy Balance

The central nervous system (CNS) is a key organ in energy balance and body weight regulation. The CNS elicits these functions through three mechanism: (a) controlling food-related behaviors, including food intake (b) controlling the autonomic nervous system, which regulate the energy expenditure and other metabolic processes; and (c) controlling the neuroendocrine system by modulating hormone secretion. The modulation and coordination of this complex systems occurs in different brain areas [10,11]. However, it is well known that the hypothalamic neuronal networks integrate peripheral information (i.e., nutrients and hormones) to modulate energy balance [12,13]. The hypothalamus, a central structure composed of anatomically distinct nuclei interconnected via axonal projections, is at large the most studied area in term of its regulation of food intake and body weight. In this neuronal network, we can distinguish the hypothalamic arcuate nucleus (ARC) as the best positioned nucleus to receive signals from the periphery and to mediate a subsequent homeostatic response to peripheral tissues. This nucleus is considered the “master hypothalamic center” for feeding control [13,14].

The ARC is composed by two antagonist neuronal populations, neuropeptide Y/Agouti related peptide (NPY/AgRP) neurons that induces a positive energy balance and proopiomelanocortin (POMC) neurons and cocaine and amphetamine related transcript (CART) that when activated induces negative energy balance. Both neuron types regulate food intake, energy expenditure and nutrient partitioning. When energy intake exceeds expenditure, the expression of orexigenic neuropeptides, such as AgRP and NPY decreases, whereas the expression of anorexigenic neuropeptides, such as CART and POMC increases [3,10]. Opposite changes occur when energy expenditure exceeds intake.

Hypothalamic neurons in the ARC respond to peripheral nutrients, such as glucose and fatty acids as well as hormones, such as leptin and ghrelin, the two most important hormones in the regulation of food intake and body weight [11,12]. Leptin, secreted by, and in proportion to white adipose tissue (WAT), informs the hypothalamus of energy buildup and activates POMC/CART neurons and inhibits NPY/AgRP neurons, resulting in an inhibition of feeding and an increase in energy expenditure. In contrast, ghrelin is a stomach-derived peptide that increases food intake through increased NPY and AgRP expression. Therefore, both leptin and ghrelin use these pathways in opposite ways to exert their roles on energy balance.

Adjacent to the ARC we find the ventromedial nucleus (VMH), a hypothalamic area that receives projections mainly from NPY/AgRP and POMC neurons. These VMH neurons express steroidogenic factor 1 (SF1) and project their axons at the same time to the ARC and to secondary hypothalamic nuclei, as well as regions of the brain stem [15]. These neurons are activated in respond to changes in circulating leptin and other metabolic signals, and due to their projections to the ARC, an elaborate neuronal circuitry underlying metabolic homeostasis is formed [15,16].

Another relevant metabolic nucleus is the lateral hypothalamic area (LHA), a region of the brain that has for long been known for its role in the regulation of eating and considered the most interconnected area of the hypothalamus. This hypothalamic area is mainly composed of two types orexigenic neuronal populations, namely orexin-producing neurons and melanin-concentrating hormone (MCH)-producing neurons [17,18,19].

Taking into account that several mechanisms are involved in the development and maintenance of obesity within the hypothalamus, in the next section we introduce the role of a highly conserved family of class III deacetylases denominated Sirtuins, (SIRTs) as one of these relevant metabolic regulators.

## 2. Sirtuins: Metabolic Sensors

Mammalian SIRTs are homologous to the *Sir2* gene of *Saccharomyces cerevisiae*. Seven *Sir2* orthologues, members of the SIRT family (SIRT1-7), have been described in mammals. All members share the conserved SIRT domain, but they differ in subcellular location and function [20,21,22]. SIRTs actively participate in the regulation of energy metabolism related to diet/caloric restriction, but they also have an important role in other metabolic processes, such as aging, cell survival, and DNA repair [23,24].

The aim of this review is to describe the hypothalamic actions of SIRTs in a metabolic context. Until now, only SIRT1 and SIRT6 have been shown to act in the hypothalamus to control energy balance. Therefore, in this review, we will describe in detail some of the most important hypothalamic mechanisms by which these two SIRTs regulate energy metabolism.

### 2.1. Hypothalamic SIRT1 in the Control of Energy Balance

SIRT1 is expressed in a wide range of tissues and organs, such as the liver, pancreas, heart, muscle, and adipose tissue (Reviewed in [20,21,22]). SIRT1 is also expressed in important metabolic centers of the brain, including ARC, VMH, dorsomedial nucleus and paraventricular nucleus of the hypothalamus (PVN), the area postrema and the nucleus of the solitary tract in the hindbrain (For review see [20,21,22]). SIRT1 is involved in the control of metabolic processes that regulate body weight, such as control of food intake [25,26,27,28], adiposity [26,29], energy expenditure [29], thermogenesis of brown adipose tissue (BAT) and “browning” of WAT [29] (Figure 1).

With regard to genetic animal models for SIRT1, the global knock out (KO) mice for SIRT1 are not viable or have several metabolic complications [30], whereas moderate overexpression of SIRT1 in mice improves several metabolic parameters associated with obesity [31] (Figure 1). Consistently, the pharmacological activation of SIRT1 produces a variety of beneficial metabolic actions in vivo (Figure 1). In this sense, natural activators of SIRT1, such as the polyphenol resveratrol (RSV) [32,33] or synthetic molecules [34,35,36], have been used for this purpose. These pharmacological activators improve insulin sensitivity, increase insulin secretion, enhance mitochondrial function, decrease adiposity and lower glucose levels [37,38,39,40,41]. Intriguingly, a negative energy balance was also observed in mice following pharmacological inhibition of SIRT1 [26,27]. Therefore, a controversy exists on the role of SIRT1 in regulation of energy metabolism and more studies are needed to ascertain the mechanism of action of these SIRT1 modulators.

### 2.2. SIRT1 and Food Intake

Several studies have shown that pharmacological inhibition of SIRT1 in the hypothalamus reduces the caloric intake and body weight in rodents [26,27], which occur through the modulation of the central melanocortin system [25,27,28]. In fact, central administration of a melanocortin receptor antagonist (SHU9119) reversed the anorectic effect of hypothalamic SIRT1 inhibition [27]. Indeed, this anorexigenic action is mediated by the repression of AgRP and stimulation of POMC levels. In addition, SIRT1 requires the Forkhead Box O1 transcription factor (FOXO1) to exert these actions, since FOXO1 inhibition blocks the actions of SIRT1 on AgRP and POMC neurons [27]. However, SIRT1 has other downstream effectors, e.g., the actions of SIRT1 on food intake are mediated by the mitochondrial redox machinery, because the decrease in feeding induced by the action of EX-527, a pharmacological SIRT1 inhibitor, is impaired in uncoupling protein-2 KO animals [28].

In agreement with these data, hypothalamic SIRT1 can act as a mediator of several nutritional hormones (reviewed in [42]). In fact, the pharmacological blockade of SIRT1 in the CNS or the genetic inhibition of p53, a downstream effector of SIRT1, prevents the orexigenic action of ghrelin, by impairing the regulation of this hormone on its target neuropeptides NPY/AgRP and its target transcription factors namely FOXO1, cAMP response element-binding protein and brain-specific homeobox transcription factor [25]. Subsequent studies with conditional KO mice models for SIRT1 and p53 in specific neuronal populations show that the food intake induced by ghrelin rely on the action of these energy sensors specifically in AgRP neurons [28,43]. These data highlight the relevance of SIRT1/p53 pathway in the orexigenic action of ghrelin.

Consistently with these findings, it has been shown that SIRT1 mediates the metabolic actions regulated by other orexigenic hormone such as MCH. In this study, it was shown that the action of MCH on food intake, glucose homeostasis and adiposity rely on SIRT1 in POMC neurons, due to the specific deletion of SIRT1 on a subset of these neurons comprise the metabolic actions of MCH [26]. These data reveal for the first time the neuronal basis of the metabolic effects of MCH on body weight and food intake.

Another study demonstrated the importance of SIRT1 in food selection, and that SIRT1 mediates the regulation of macronutrient choice [44]. In this study, targeting central SIRT1 by gain-of function and loss-of-function mice models, it was shown that the preference for high sucrose diet was reduced when SIRT1 was stimulated at neuronal level, while it was increased when SIRT1 was inhibited at this level. These effects of diet preference over sucrose were shown to be mediated by SIRT1 specifically in oxytocin neurons, since overexpression or deletion of SIRT1 in oxytocin PVN neurons displayed the same results on sucrose preference as the neuronal manipulation of SIRT1. However, when standard diet (SD) vs. high fat diet (HFD) was offered to the animals in the latter mentioned mice model, no differences were found, which show that SIRT1 in oxytocin neurons regulates the preference for sucrose, but not for fat. The fibroblast growth hormone-21 (FGF-21) is a peripheral signal targeting oxytocin neurons to regulate macronutrient preference. In the same study SIRT1 was identified as a regulator of sucrose preference by promoting FGF21 sensitivity in oxytocin neurons.

### 2.3. SIRT1 in Hypothalamic Neurons

Studies with conditional mice models have shown that the effects of SIRT1 in the hypothalamus may depend on the neuronal type in which it acts (Figure 2).

It has been reported that inhibition of SIRT1 specifically in POMC neurons does not significantly alter body weight or adiposity in SD-fed mice [29]. However, these mice are more vulnerable to diet-induced obesity (DIO) due to a reduction in energy expenditure and a reorganization of WAT through the sympathetic nervous system. Consistent with these data, overexpression of SIRT1 in POMC neurons has the opposite effect, and these mice showed a lean phenotype due to an increase in energy expenditure and activation of sympathetic nervous system in adipose tissue [45].

Additionally, SIRT1 is essential for the metabolic functions of AgRP neurons. The inhibition of SIRT1 specifically in AgRP neurons decreases the consumption of food, which leads to a decrease in adiposity and consequently a reduced body weight [28]. There are, however, also opposing experimental data, demonstrating that overexpression of SIRT1 in AgRP neurons improves weight gain and reduces the food intake associated with age in mice [45].

Overexpression of SIRT1 in SF1 neurons shows protective effects against DIO, with reduction in body weight, fat mass, and leptin levels, and increased energy expenditure [46]. Accordingly, the specific inhibition of SIRT1 in SF1 neurons makes HFD fed mice more susceptible to obesity and diabetes mellitus [46].

These data indicate that SIRT1 plays an essential role in hypothalamic neurons that are associated with the metabolic adaptations that determine body weight gain in obesity.

### 2.4. SIRT1 a Link between Metabolism and Reproduction

SIRT1 has been shown to be the link between metabolism and reproduction through hypothalamic neurons [47]. First, it has been observed how female animals that overexpress SIRT1 have a normal body weight and show a delay in the development of puberty. Using virogenetic techniques, it was demonstrated that the overexpression of SIRT1 specifically in the ARC promoted this delay. Moreover, in this work it was also shown that SIRT1 is expressed in kisspeptin, neurokinin B and dynorphin (KNDy) neurons, which express the *Kiss1* pubertal activator gene. It is important to highlight that the content of SIRT1 in the KNDy neurons of the ARC and the recruitment of SIRT1 towards the *Kiss1* promoter in vivo change depending on the nutritional status of the animal. In states of overnutrition, the neuronal content of SIRT1 and its association with *Kiss1* decrease, which accelerates and promotes puberty [47]. In contrast, malnutrition raises SIRT1 levels, prolongs the repression of *Kiss1* by SIRT1, and delays puberty. These results identify SIRT1-mediated *Kiss1* inhibition as a key epigenetic mechanism by which nutritional cues and obesity influence puberty in mammals.

In summary, hypothalamic SIRT1 signaling appears to be a key mediator of energy metabolism and the physiological response to obesity.

## 3. Hypothalamic SIRT6 in the Control of Energy Balance

SIRT6 is a nuclear protein associated with physiological and pathological processes, regulating obesity, insulin resistance, inflammation and energy metabolism [48]. It is expressed in various tissues, with the highest level of expression in adipose tissue, skeletal muscle and heart [49,50,51]. SIRT6 is also highly expressed in the CNS, and its expression is regulated by the availability of nutrients, showing low levels in the hypothalamus and specifically in POMC neurons in obesity [52]. Therefore, SIRT6 is postulated as a relevant energy sensor at the central level and a promising pharmacological target in the regulation of energy metabolism.

A seminal work in the study of SIRT6 described that global SIRT6 KO mice suffer a severe multisystemic phenotype, with severe hypoglycemia and a short life expectancy [49]. A remarkable characteristic of these mice is that they show a delay in postnatal growth associated with low levels of insulin-like growth factor 1. In agreement with these results, mice in which SIRT6 was inactivated at the neural level (NSIRT6-KO) display a significantly reduced growth during development, smaller adult phenotype and lower body weight compared to controls [50]. These results demonstrate that neural inactivation of SIRT6 is sufficient to cause growth retardation. However, elderly male NSIRT6 KO mice, but not female mice, were significantly heavier than controls [50].

Mice deficient for SIRT6, specifically in POMC neurons (SIRT6 POMC KO), showed no changes in body size or differences in glucose levels [52]. Therefore, hypoglycemia in SIRT6 KO mice and growth retardation in NSIRT6 KO mice were not due to SIRT6 deficiency in POMC neurons. On the contrary, SIRT6 POMC KO mice showed a modest, but significant, increase in body weight under SD that was greater when the mice were fed HFD [52]. This increase in body weight is associated with a higher adiposity in WAT and liver, caloric intake, glucose intolerance and less energy expenditure. Consistently, it was also found that the activity of the leptin signaling pathway was significantly reduced in SIRT6 POMC KO mice, showing high levels of leptin in the serum of these mice both under SD and HFD [52].

Interestingly, and in line with these previous results the global overexpression of SIRT6 decrease body weight, adiposity and glucose tolerance [51]. Moreover, and consistent with these data, the study of Tang et al. has demonstrated that overexpression of SIRT6 in the ARC of obese mice decreases body weight, adiposity and food intake [52]. These data suggest that the activation of SIRT6 in POMC neurons could be the responsible of this effect, however this possibility was not tested yet.

Altogether, these studies postulate to SIRT6 as an essential player in the regulation of body energy homeostasis.

## 4. Therapeutic Options

The positive effects of SIRTs in animal models on prevention and treatment of obesity and metabolic syndrome suggested their use as therapeutic agents. However, the knowledge about the involvement of SIRTs on clinical trials for the treatment of metabolic disease is limited.

In this sense, the administration of RSV, the most studied natural SIRT1 activator, reduces adipocyte size, blood glucose, preserves pancreatic β-cells and improves insulin action in rhesus monkeys fed a high-fat, high sugar diet for 2 years [53,54]. Importantly, treatment with RSV during 30 days in obese men confirmed these initial reports in non-human primates [55]. Other epidemiological studies in humans showed positive effects of RSV in patients with diabetes or obesity-mediated insulin resistance [40,56,57]. Accordingly, RSV improves plasma triglyceride concentration [37,41], lowers circulating cytokine levels [37,38,39], elicits better metabolic flexibility with lower homeostatic model assessment for insulin resistance index [37], suppresses postprandial glucagon responses, decreases resting metabolic rate and improves respiratory quotient in obese subjects [37,58]. Interestingly, all of these effects elicited by RSV on energy balance, glucose homeostasis and insulin sensitivity were not observed in non-obese healthy humans [59].

It is important to highlight that in spite of these results, the use of the natural activator RVS as a therapeutic target bear some controversies. For example, the use of RVS shows low tolerability and specificity, inconsistent dosages activity and interactions with other compounds (for a discussion of this issue see [60]). These shortcomings have stimulated to the development of a number of new small synthetic molecules activators. However, although these molecules are 1000 times more potent and their bioavailability are improved compared to the natural compounds, they do not show a superior efficacy in preclinical or clinical studies [34,35,36]. For example, some epidemiological observations with the synthetic SIRT1 activator, SRT2104, showed that the administration of this compound during 28 days produces a mild reduction in the glucose levels and slightly improvement in the body weight [61]. Other studies in elderly or type 2 diabetic patients shows that the administration of SRT2104 reduces the lipid profile and the body weight [62] without changes in glucose levels [36,61].

Despite these controversies, SIRT1 activators often show an acceptable efficacy in treating metabolic disturbances in clinical trials, where improvements are restricted to subjects with a pathological condition. Thus, more studies with larger cohorts are necessary to obtain a consensus for the actions, specificity, dosage, and effectiveness of these SIRT1 activators.

## 5. Conclusions and Remarks

In this review we cover the most relevant studies focusing on the metabolic actions of SIRT1 and SIRT6 at the hypothalamic level. In this sense, it is important to highlight that the effect of hypothalamic SIRT1 on energy balance depends on the neuronal type where it acts, which has been demonstrated for a variety of metabolic effects. E.g., the specific deletion of SIRT1 in POMC and SF1 neurons from HFD-fed mice shows that SIRT1 regulates metabolism through energy expenditure but does not affect food intake, whereas specific modulation of SIRT1 in AgRP neurons specifically affects eating behavior. Other examples of the relevance of this deacetylase in the regulation of metabolic processes came from studies that describe its involvement in the signaling pathways of some relevant metabolic hormones. More specifically, it has recently been shown that SIRT1, in addition to mediate the orexigenic action of ghrelin, is essential for the metabolic actions of MCH. Importantly in both cases, the ARC melanocortin system plays a fundamental role in the action of SIRT1 in these processes.

SIRT1 may also regulate diet preference via actions in the PVN Oxitocin neurons. This is relevant since the effectiveness of the diet and its impact on body weight depends both on the amount of caloric intake that is consumed and the balance of macronutrients of the ingested meal. It is also important to highlight the effects of SIRT1 on conditions related to obesity, such as puberty. SIRT1 has been shown to be a key factor in the development of puberty. Specifically, it has been identified that the ARC is the main integrator area of these effects, with epigenetic inhibition of *Kiss1* by SIRT1 as a key mechanism by which nutritional signals and obesity influence puberty of children.

In addition to the effects of SIRT1, this review also provides a comprehensive overview on the involvement of hypothalamic SIRT6 in the control of energy metabolism. It has been shown recently that the specific deletion of SIRT6 in POMC neurons increases food intake and decreases energy expenditure, promoting an obesogenic phenotype in mice. These results suggest that SIRT6 has a protective effect against DIO by its action in POMC neurons. In this sense, future studies should focus on the possible effects of SIRT6 modulation in other hypothalamic neurons such as AgRP or SF1 neurons. Therefore, all these results strongly support SIRT6 as a new hypothalamic molecular mediator in the regulation of energy homeostasis.

The data described in this review reveal SIRT1 and SIRT6 as multifaceted mediators of energy metabolism, affecting processes such as food intake, food preference, puberty, body weight, adiposity, glucose homeostasis and insulin resistance.

Apart from these considerations there are controversial results regarding the central actions of these SIRTs that we have discussed. While it seems clear that SIRT1 activators improve several metabolic parameters, in some circumstances this is also true for SIRT1 inhibitors. For example, neural deletion of SIRT1 improves insulin sensitivity and glucose intolerance in mice, while the administration of the pharmacological inhibitor EX-527 improves body weight, food intake, hepatic steatosis and fibrosis in diabetic rats [25,26,27,63]. In this sense, the healthy phenotype observed in these rodent models following SIRT1 inhibition may at first seem contradictory, but there are other previous examples in the literature involving other mammalian SIRTs that show similar results. Indeed, SIRT6 overexpression protected against DIO [51] in the same way that SIRT6 deletion did [49]. There is currently no clear explanation for the controversial results regarding the central actions of these SIRTs, but it is believed that the processes that involves metabolic regulation are highly complex and that this may sometimes lead to counterintuitive findings. Therefore, these data imply that the implications for these variety of effects are that their levels must be under a fine-tuning regulation and that their alterations may lead to unpredictable side effects without an evident dose-response relationship.

In spite of these somewhat contradictory results, we believe that SIRT1 and SIRT6 emerge as important targets for the development of novel therapies in the control of obesity and its comorbidities. Importantly, it should be noted that the fact that these proteins regulate energy balance in specific high-calorie regimens is of paramount importance for the development of new strategies to combat metabolic diseases. Finally, we should not forget the potential involvement of the other SIRTs different from SIRT1 and SIRT6 in the control of energy balance at the hypothalamic level, which should be the focus of future studies.

## Figures and Tables

**Figure 1 ijms-22-01430-f001:**
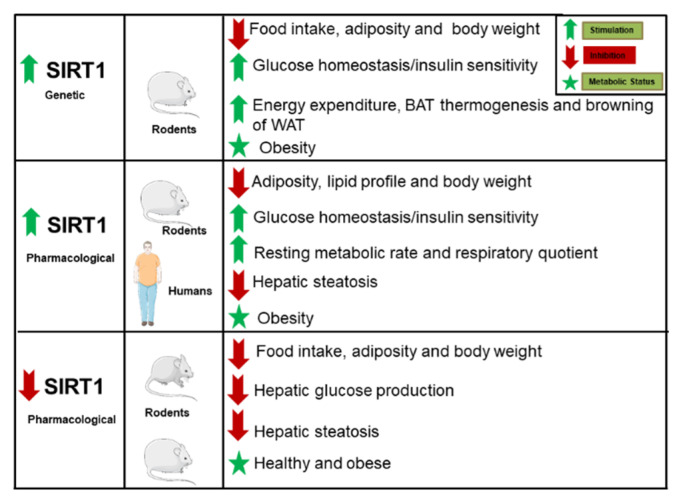
Description of the metabolic actions elicited by SIRT1 in mice and humans. Pharmacological or genetic modulation of SIRT1 regulates body weight, adiposity in WAT and liver, food intake, energy expenditure, resting metabolic rate, respiratory quotient, BAT thermogenesis, browning of WAT, glucose homeostasis and insulin sensitivity. green arrow: stimulation, red arrow: inhibition, green star:metabolic state. Abbreviations used-BAT; brown adipose tissue; SIRT1: sirtuin 1; WAT; white adipose tissue. The figures were generated by using materials from Servier Medical Art (Servier) under consideration of a Creative Commons Attribution 3.0 Unported License.

**Figure 2 ijms-22-01430-f002:**
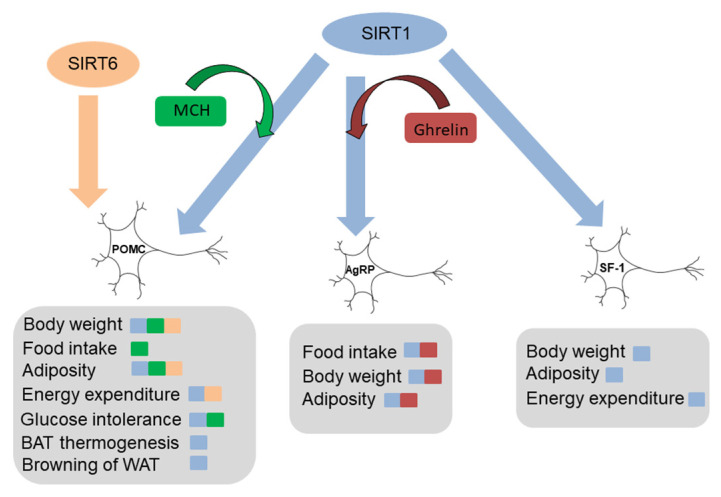
SIRT1 and SIRT6 in hypothalamic neurons: The effects of SIRT1 and SIRT6 in the hypothalamus may depend on the neuronal type. Abbreviations used-AgRP: Agouti-related protein neurons; BAT; brown adipose tissue; MCH: melanin concentrating hormone; POMC: Proopiomelanocortin neurons; SF-1: steroidogenic factor 1; WAT; white adipose tissue. The figures were generated by using materials from Servier Medical Art (Servier) under consideration of a Creative Commons Attribution 3.0 Unported License.

## Data Availability

Not applicable.

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
