# Peer review of "Hypothalamic Actions of SIRT1 and SIRT6 on Energy Balance"

_ijms, 2021, doi:10.3390/ijms22031430_

Round 1
Reviewer 1 Report
The introduction section for obesity needs to be improved, please!!!
If it is well known, why speak about anatomically ???
You already started the paragraph with the word anatomically.
Please re-phrased!!
Describe ghrelin and leptin before you launch them in the text.
Describe expenditure, basal metabolism, etc. please and briefly.
Chapter 2 and 3 are clear.
Dear Authors, your review is very hard to read, the first part is uncompressible, gets better within the 2nd and 3rd chapter but after that you get the impression of atrial flutter on ECG strip. I believe information’s and observations are just through there with no context.
Author Response
Referee comments to author:
Referee 1:
The introduction section for obesity needs to be improved, please!!!
RESPONSE: We agree with the reviewer that the introduction could be improved, and we have now rewritten and reorganized the introduction section following the reviewer’s recommendations. We believe that the introduction is now more focused and easier to read.
If it is well known, why speak about anatomically ???
You already started the paragraph with the word anatomically.
Please re-phrased!!
RESPONSE: We agree with these comments, and as we pointed before we have changed some paragraphs of the introduction section and now this specific paragraph was deleted.
RESPONSE: Following the reviewer’s recommendation, we have included the following modifications:
Describe expenditure, basal metabolism, etc. please and briefly.
Revised text, section 1.1, line 30:
Obesity is defined as a state on excess of fat produced as a consequence of a high energy intake and a low energy expenditure that leads to serious health problems [3,4]. While food intake is tightly regulated by homeostatic and hedonic drives [5], changes in energy expenditure can be explained by differences in basal metabolism (energy expended to keep the body functioning at rest), physical activity and adaptive thermogenesis [6]
Describe ghrelin and leptin before you launch them in the text.
Revised text, section 1.2, line 69:
Hypothalamic neurons in the ARC respond to peripheral nutrients, such as glucose and fatty acids as well as hormones, such as leptin and ghrelin, the two most important hormones in the regulation of food intake and body weight[11,12]. Leptin, secreted by, and in proportion to white adipose tissue (WAT), informs the hypothalamus of energy buildup and activates POMC/CART neurons and inhibits NPY/AgRP neurons, resulting in an inhibition of feeding and an increase in energy expenditure. In contrast, ghrelin is a stomach-derived peptide that increases food intake through increased NPY and AgRP expression. Therefore, both leptin and ghrelin use these pathways in opposite ways to exert their roles on energy balance.
Chapter 2 and 3 are clear.
We thank the Reviewer for this positive and encouraging comment on these sections of the manuscript.
Dear Authors, your review is very hard to read, the first part is uncompressible, gets better within the 2nd and 3rd chapter but after that you get the impression of atrial flutter on ECG strip. I believe information’s and observations are just through there with no context.
RESPONSE: We thank the reviewer for the clear feedback, which we to a large extent agree upon. We have now rewritten the text in the introduction part considerably. In addition, we made some corrections in the sections 4 and 5, where we have rephrased some sentences for a better readability and flow.
Reviewer 2 Report
In this article, the author reviewed the metabolic functions of the SIRT1 and SIRT6 in the hypothalamus. Both of SIRT1 and SIRT6 control the energy balance in the hypothalamus. However, there are some shortcomings and questions.
- In the line 16: central role of these two sirtuins have been shown to be crucial in eliciting responses against met-, the “have” should be “has”.
- In the line 126: Until date should be “Until now” or “To date”.
- In the line 236: in the ARC promotes this delay, the “promotes” should be “promoted”.

Author Response
Referee comments to author:
Referee 2:
In this article, the author reviewed the metabolic functions of the SIRT1 and SIRT6 in the hypothalamus. Both of SIRT1 and SIRT6 control the energy balance in the hypothalamus. However, there are some shortcomings and questions.
- In the line 16: central role of these two sirtuins have been shown to be crucial in eliciting responses against met-, the “have” should be “has”.
- In the line 126: Until date should be “Until now” or “To date”.
- In the line 236: in the ARC promotes this delay, the “promotes” should be “promoted”.
RESPONSE: We would like to thank the Reviewer for his/her comments and we apologize for these mistakes, which are corrected in the revised manuscript